# The Influence of News Consumption Habits and Dispositional Traits on Trust in Medical Scientists

**DOI:** 10.3390/ijerph20105842

**Published:** 2023-05-17

**Authors:** Meng Zhen Larsen, Michael R. Haupt, Tiana McMann, Raphael E. Cuomo, Tim K. Mackey

**Affiliations:** 1Global Health Policy and Data Institute, San Diego, CA 92123, USA; 2S-3 Research LLC, San Diego, CA 92123, USA; 3Department of Cognitive Science, University of California, San Diego, CA 92093, USA; 4Global Health Program, Department of Anthropology, University of California, San Diego, CA 92093, USA; 5Department of Anesthesiology, School of Medicine, University of California, San Diego, CA 94720, USA

**Keywords:** news consumption habits, dispositional traits, trust in medical institutions, misinformation, public health

## Abstract

Public trust in medical institutions is essential for ensuring compliance with medical directives. However, the politicization of public health issues and the polarized nature of major news outlets suggest that partisanship and news consumption habits can influence medical trust. This study employed a survey with 858 participants and used regression analysis to assesses how news consumption habits and information assessment traits (IATs) influence trust in medical scientists. IATs included were conscientiousness, openness, need for cognitive closure (NFCC), and cognitive reflective thinking (CRT). News sources were classified on the basis of factuality and political bias. Initially, readership of liberally biased news was positively associated with medical trust (*p* < 0.05). However, this association disappeared when controlling for the news source’s factuality (*p* = 0.28), while CRT (*p* < 0.05) was positively associated with medical trust. When controlling for conservatively biased news sources, factuality of the news source (*p* < 0.05) and NFCC (*p* < 0.05) were positively associated with medical trust. While partisan media bias may influence medical trust, these results suggest that those who have higher abilities to assess information and who prefer more credible news sources have a greater trust in medical scientists.

## 1. Introduction

### 1.1. Mistrust in Medical Institutions and Misinformation Susceptibility

Public trust in medical institutions is critical for ensuring compliance with public health directives, especially during health crises. However, the rapid dissemination of misinformation has become a critical public health threat and a source of increased medical mistrust [1]. Medical mistrust refers to suspicion or lack of trust in medical organizations [2,3]. Prior to the COVID-19 pandemic, public trust in medical scientists had been complicated and influenced by a number of factors such as race and an individual’s past negative healthcare experiences [4,5,6,7]. Additionally, mistrust in medical institutions prior to COVID-19 has been linked to decreased treatment adherence and utilization of healthcare services, dissatisfaction with medical care, and decreased engagement with biomedical research, including vaccine trials [3,7,8,9]. Furthermore, medical mistrust has historically resulted in vaccine hesitancy among parents, which has led to outbreaks of vaccine-preventable diseases and, consequently, preventable mortality [8,9]. Hence, medical mistrust is not a novel phenomenon within the United States.

Antivaccine and vaccine hesitancy beliefs possess a longstanding presence in the American public sphere. However, antivax beliefs have not been confined to specific partisan lines. Antivax beliefs were prominent among white enclaves in the 1970s during the neoliberal counterculture movement [10]. Previous research characterized attributes of individuals who hold antivax beliefs, but this research found that antivax beliefs are not the result of political ideology alone [11]. Moreover, increases in antiestablishment sentiment as part of conservative political discourse in the United States have featured anti-science and antivax dialogue, as individuals challenge the legitimacy of public institutions [12]. Therefore, antivax beliefs have proven difficult to confine to a specific group due to the history of antivax beliefs crossing political lines and the influence of non-political factors on vaccine attitudes.

Yet, understanding medical mistrust has become increasingly paramount during the COVID-19 pandemic. The urgent need to promote vaccination of individuals to enable an effective public health response against the virus has been complicated by the rapid dissemination of misinformation [13,14], and medical mistrust has been identified as a key component to vaccine hesitancy [15]. Mistrust and misinformation have contributed to numerous conspiracy theories which may harm public health in other ways, including questioning motivations behind efforts to mitigate the spread of COVID-19 (e.g., masking and social distancing) [16]. Additionally, ongoing structural racism in healthcare is a driver of medical mistrust in the Black community within the US. This has contributed to vaccine hesitancy in the Black community and has exacerbated existing health disparities [17].

Critically, the COVID-19 pandemic has introduced a new wave of antivax and anti-science beliefs, resulting in the need to characterize individuals who do not trust medical scientists. By examining how factors such as news consumption habits and dispositional traits relate to trust in medical scientists, it is possible to develop more effective counter-campaigns that address misinformation spread for health-related topics such as vaccination uptake. Therefore, to better understand misinformation spread and medical mistrust during critical public health crises, this study adds to the extant literature by identifying how information assessment traits and news consumption habits such as bias and factuality of content influence medical trust. To achieve this, this study classified news sources and deployed a survey to respondents who were recruited from Amazon Mechanical Turk. We measured preference of news sources, as well as information assessment traits such as conscientiousness, need for cognitive closure (NFCC), and cognitive reflective thinking (CRT), to better understand factors which influence medical trust.

### 1.2. The Influence of Media Outlets on Misinformation Susceptibility

Classifying misinformation and the behaviors of those likely to spread, engage with, and trust such information is a key prevention measure against uncertainty amidst a public health crisis [18,19]. Due to today’s media landscape being complex and ever-evolving with a multitude of available news sources [20], it is likely that the general public can feel overwhelmed and confused with access to almost limitless options that range widely in political bias and factuality [21]. In the current study, we define news consumption habits as one’s individual preference for established media outlets.

The media outlets people choose as news sources can have a major impact on the stories and information that they are exposed to, which can subsequently influence their perceptions of public issues. For example, one study found that, despite low crime rates in Ireland, biased media reporting that exaggerates extreme offences has contributed to the public misperception that Ireland has a law-and-order crisis [22]. While media outlets make attempts to correct misinformation by reporting on false stories, these efforts may further exacerbate their spread since a story is more likely to receive exposure from a larger proportion of the population when reported by mainstream media outlets as opposed to relatively niche websites [23]. The reporting of false stories by mainstream news outlets can also contribute to the illusory truth effect, where multiple studies show that repeated exposure to the same information is positively associated with perceived truthfulness [24,25], and this effect remains for statements and headlines containing misinformation [26,27]. Overall, the ability of news outlets to choose what stories to report on and how often they are covered can influence public perceptions of emerging and ongoing health crises, as well as opinions of medical authorities and institutions addressing these issues. Since the stories a news source reports on can be greatly impacted by its political leanings, this implies that the political biases associated with media outlets are also important factors for examining public perceptions of medical authorities.

Existing research has found that political ideology and information assessment traits may influence news consumption habits and the dissemination of misinformation. For example, recent studies showed a positive association between conservative ideology and an individual’s susceptibility to COVID-19 misinformation [28], and that conservatives tend to gravitate toward less factual news sites than liberals [29]. However, existing evidence indicates that information assessment ability may override the effect of political beliefs on misinformation susceptibility. While a recent study found that liberal ideology predicted a lower tendency to share fake news, controlling for conscientiousness resulted in a nonsignificant effect of ideology on an individual’s tendency to spread misinformation [30]. Furthermore, past research found that preference for conservative-leaning news sources is correlated with conspiracy ideation and increased belief in COVID-19 conspiracies [31]. On the basis of these previously reported findings, we hypothesize the following:

**H_1_.** 
*Those with a higher preference for factual news sources are more likely to trust medical scientists.*


**H_2A_.** 
*Preference for liberal instead of conservative news sources is associated with higher medical trust.*


**H_2B_.** 
*A preference for factual news sources and one’s dispositional traits will disrupt the influence of partisan bias.*


### 1.3. Overview of Tested Information Assessment Traits on Trust in Medical Institutions

Previous work has examined the association between information assessment traits and susceptibility to health-related misinformation. A widely used metric in the literature that measures one’s propensity to engage in analytic thinking is the cognitive reflection test (CRT), which administers word problems in which the answer that comes “first to mind” is wrong and the correct answer requires one to pause and reflect carefully [32,33]. Existing research has found that lower CRT is correlated with a lower ability to identify fake news [34] and higher likelihood to share low-credibility news sources on Twitter [35]. Since COVID-19 misinformation consistently opposes public health institutions [36], this suggests that CRT could also influence one’s trust in medical institutions.

Need for cognitive closure (NFCC) is an information trait which gauges someone’s desire for order and consistency, as well as someone’s uneasiness with ambiguity [37]. Previous research showed that NFCC does not directly moderate an individual’s susceptibility for believing misinformation after multiple exposures [38]. However, other studies found that NFCC increases the spread of misinformation on social media because conspiracists exhibit avoidance behavior toward scrutinizing evidence of claims [39]. Due to COVID-19 misinformation being so abundant and widespread [36], this could cause public health information to be ambiguous at times. Since NFCC measures one’s comfort with ambiguity, and since individuals with high NFCC express avoidance behavior when asked to support their beliefs with evidence [40], past research suggests that NFCC could increase one’s susceptibility to misinformation which subsequently can influence trust in medical institutions as well.

The conscientiousness trait from the Big Five Inventory (BFI) is intended to capture an individual’s inclination to be methodical, careful, and goal-oriented [41,42,43]. Past research found that possessing low levels of conscientiousness is associated with a higher tendency to share fake news [30]. Conversely, high levels of conscientiousness are negatively correlated with heuristic processing which is associated with a tendency to spread misinformation [44]. Therefore, due to the dissemination of misinformation decreasing people’s willingness to abide by public health guidelines [13,14], we chose to use conscientiousness for the current study to explore how this trait could influence trust in medical institutions.

Individuals with high levels of BFI openness are open to new ideas; however, there are discrepancies concerning its influence on misinformation susceptibility. Some research found that high levels of openness were positively correlated with the ability to discern fake news [45], while other work found that openness did not have a significant influence on one’s ability to classify false news [46]. With past research showing conflicting results on the ability of openness to influence the discernment of fake news, the current study examines how this trait can possibly influence medical trust.

On the basis of the previously reported findings, we hypothesize the following:

**H_3a_.** 
*Those with higher CRT are more likely to trust medical scientists.*


**H_3b_.** 
*Those with higher NFCC are less likely to trust medical scientists.*


**H_3c_.** 
*Those with higher conscientiousness are more likely to trust medical scientists.*


**H_3d_.** 
*Those with higher openness are more likely to trust medical scientists.*


## 2. Materials and Methods

This study was based on a secondary analysis of survey data examining how dispositional traits influence engagement of misinformation on social media. The primary analysis will be reported in a separate paper. A total of 858 respondents were recruited from Amazon Mechanical Turk (MTurk) on 22 September 2021, according to whether they stated that they had a Twitter account. To be included in the analysis, individuals must have possessed a survey completion time over the bottom 90th percentile (>12 min, median survey completion time = 26.85 min) and successfully answered an attention check question. Out of 1000 total respondents, the previous screening methods finalized the sample size to 858 respondents. Respondents were monetarily compensated according to standard survey-taking rates on the platform. Ethics approval was obtained from the Human Research Protections Program at the University of California, San Diego.

Two eight-item subscales from the Big Five Inventory [41,42,43] were used to evaluate conscientiousness and openness levels for each participant. A 15-item subscale was used to evaluate a participant’s need for cognitive closure (NFCC) [47]. In order to measure one’s tendency to engage in reflective thinking, we used the cognitive reflection test (CRT), which asks questions that require deliberation in order to respond correctly [32]. For example, one question asked “A bat and a ball cost $1.10 in total. The bat costs $1.00 more than the ball. How much does the ball cost?” To measure CRT and invoke the necessity for deliberation, the intuitive answer is $0.10; however, the correct answer is $0.05.

In order to measure news consumption habits, participants were asked to select which news sources they preferred to stay up to date on current events among 23 mainstream outlets (e.g., CNN, FOX, and MSNBC; see Table 1). Trust in medical scientists was measured using a four-item scale adapted from the 2019 Pew Research Center’s American Trends Panel survey [48], which asked respondents “How much confidence, if any, do you have in each of the following to act in the best interests of the public?” with “medical scientists” as one of the queried subjects. Response options ranged from 1 = “no confidence at all” to 4 = “a great deal”. Medical trust was measured as a single-item measurement.

Additionally, the website Media Bias/Fact Check (MBFC) was used to assign political bias and factuality levels to each news source. MBFC is an independent website that was founded in 2015, which receives funding from reader donations and third-party advertising. To determine the political bias and factuality of a news source, MBFC takes into consideration four different main categories: wording/headlines, factual/sourcing, story choices, and political affiliation. MBFC additionally takes into consideration subcategories such as bias by omission, bias by selection of sources, and loaded language use. Since its launch, MBFC has been referenced by major media outlets, universities, high schools, and libraries across the United States [49,50]. Specifically, MBFC has been referenced in research predicting factuality of news sources [51] and research measuring misinformation on social media [39].

Factuality of news sources was measured using a four-level unipolar scale: mixed factuality, mostly factual, high factuality, and very high factuality. The political bias of each news source was measured through a five-level bipolar scale: extreme-right, right-center, least-biased, left-center, and left (Table 1).

Factuality and bias media scores were calculated based on the percentage of factual, mixed, right-biased, and left-biased media outlets selected by respondents as news sources they turn to most to stay informed about current events. Due to the very high factuality category only containing one news source, this category was collapsed with the high factuality category. A percentage-based scoring was preferred over a tally score to better account for instances where a respondent listed a large number of news sources, resulting in over-representation of news sources of a certain category. To illustrate percentage-based scoring, if someone listed that they preferred receiving news from MSNBC, NYT, ABC, and PBS, their factual news score would be 0.75. On the other hand, if someone listed that they preferred receiving news from CNN and ABC, their factual news score would be 0.50.

Utilizing the factual news percentage and political bias percentage scores, ordinal logistic regression was used to assess how information assessment traits and news consumption habits are associated with medical trust. When controlling for confounding variables, the controlled variable was held constant above that specific variable’s third quartile threshold. We chose this method to control for confounding variables to see how high levels of specific news consumption habits influenced how the other independent variables were associated with an individual’s medical trust. Additionally, we used an effects plot from the “effects” R package to visualize how factual news consumption influences medical trust for each trust level. All analysis was conducted in R version 4.2.1.

## 3. Results

Out of 858 respondents, 5.94% (*n* = 51) indicated 1 = “no confidence at all” in medical scientists, 19.11% (*n* = 164) indicated 2 = “not too much” confidence, 33.33% (*n* = 286) reported 3 = “a fair amount” of trust, and 41.61% (*n* = 357) reported 4 = “a great deal” of confidence in medical scientists. Table 2 provides descriptive statistics of tested variables and sample demographics. Out of 858 participants, 5.36% (*n* = 46) reported not selecting any factual news sources. On average, out of the total news sources that respondents preferred reading, 50.52% (SD = 20.68%) were factual news sources while 28.81% (SD = 18.74%) were derived from mixed (low factual) news sources. Moreover, the average individual who reported “no confidence at all” in medical scientists indicated that 42.79% of the news sources they preferred were factual news sources and 33.05% were mixed factual news. Conversely, the average individual who reported the highest confidence in medical scientists indicated that of 53.67% of the news sources they preferred were factual news sources and 26.75% were mixed factual news. On average, respondents who reported “no confidence at all” in medical scientists indicated that they prefer less factual news than respondents who reported trusting medical scientists the most (*p* < 0.01). On average, 18.58% (SD = 15.37%) of news people preferred was liberally biased with a median of 17.39%, while, on average, 18.44% (SD = 16.10%) of news respondents preferred possessed a conservative bias with a median of 16.67%.

As illustrated in Table 3, a preliminary ordinal logistic regression found that preference for factual news sources (*p* < 0.001), preference for liberal news sources (*p* < 0.05), CRT (*p* < 0.01), NFCC (*p* < 0.05), and openness (*p* < 0.01) were positively associated with medical trust. Additionally, preference for conservative news sources and conscientiousness were not correlated with medical trust levels when controlled for the other tested variables. These results suggest that preferring factual news sources and liberal news sources increase the likelihood of trusting medical scientists. However, when controlling for factuality, the relationship of a preference for liberally biased news sources and medical trust disappeared (*p* = 0.2782; Table 4). When controlling for factuality, higher conscientiousness levels were positively associated with medical trust (*p* < 0.001; Table 4), where a one-unit increase in conscientiousness was associated with an odds ratio of 1.836.

As seen in Table 4, among a preference for liberal news sources, CRT (OR = 1.332, *p* < 0.05), and factual news consumption (OR = 5.448, *p* < 0.01) were positively associated with trust in medical scientists when controlling for other information assessment traits. Conscientiousness, NFCC, and openness were not associated with medical trust among those who highly preferred liberal news sources.

Table 4 illustrates that, among a preference for conservative news sources, preference for highly factual news sources (OR = 4.9825, *p* < 0.05) and high levels of NFCC (OR = 1.4831, *p* < 0.05) were positively correlated with trust in medical scientists after controlling for other factors. On the other hand, conscientiousness, CRT, and openness did not influence medical trust among those who preferred conservative news sources.

Table 4 outlines how information assessment traits and news consumption habits influence medical trust among Black respondents. Factuality of news sources preferred was the only statistically significant independent variable observed (OR = 19.7645, *p* < 0.05). Additionally, preference for partisan news sources and information assessment traits were not statistically significant in influencing medical trust among Black respondents.

Figure 1 is an illustration of how high levels of preference for factual news sources increase the likelihood of classification into trusting medical scientists “a great deal” (medical trust score of four). However, high levels of factuality decreased the likelihood of classification into lower levels of medical trust. These results suggest that partisan bias is not a good indicator of medical trust while those with a preference for factual news sources and higher abilities to assess information are more likely to trust medical scientists. As shown in Table 5, conscientiousness reported having an odds ratio (OR) of 1.599 while higher preference for factual news sources possessed an OR of 3.526. This implies that an individual’s conscientiousness and preference of credible news sources are positively associated with trust in medical scientists.

## 4. Discussion

While past research focused primarily on group identity when assessing trust in medical institutions, profiles incorporating news consumption habits and dispositional traits such as conscientiousness, openness, NFCC, and CRT can generate additional insights, as demonstrated in the current study. Results from this study suggest that a preference for factual news and one’s conscientiousness play a larger role than media political bias in influencing medical trust. This evidence appears to suggest that, despite some observed associations with political slant, anti-science sentiment crosses partisan lines.

We observed that the factuality of what news sources an individual prefers had a strong influence on medical trust levels (regardless of the political bias of news sources). This is consistent with past research suggesting that misinformation spread contributes to conspiracy ideation that questions public health authority and guidelines [16]. These findings are in accordance with H_1_, which stated that those with a higher preference for factual news sources are more likely to trust medical scientists. Moreover, this study’s results regarding conscientiousness, partisan bias, and medical trust are consistent with past research within the misinformation literature. Past research found that, although liberal ideology can predict a lower tendency to spread fake news, increasing conscientiousness levels erases the effect of partisan ideology on one’s tendency to spread misinformation [30]. Therefore, these findings support H_2A_ and H_2B_: although a liberal slant in media political bias was initially associated with medical trust, conscientiousness disrupted the association between media political bias and medical trust. However, we found that openness was not statistically associated with medical trust. This result is consistent with findings from Sindermann et al., who found that openness did not influence the ability to discern fake news [46], and is not consistent with H_3d_ because openness is not associated with medical trust.

Furthermore, among those who prefer liberally biased news, CRT was positively correlated with medical trust; however, among those who prefer conservatively biased news, CRT did not statistically influence medical trust. This is consistent with reported findings in the literature showing that CRT is positively correlated with public health literacy [52]. Moreover, even though CRT can influence public health literacy, the fact that conservatives have been shown to generally gravitate more toward less factual news sources than liberals [29], and that credibility of news sources is a large influencer of medical trust, this tendency toward less factual news sources may disrupt CRT’s influence on public health literacy. Our findings are consistent with H_3a_ because CRT increased medical trust among those with a preference for liberal news sources.

Additionally, need for cognitive closure (NFCC) and preference for factual news sources increased medical trust among those who preferred conservative news sources, while NFCC did not significantly influence medical trust among those who preferred liberal news sources. This is consistent with previous studies showing that conservatives value NFCC more than liberals [53], and that NFCC is positively associated with adjusting one’s misconceptions after being corrected [54]. Furthermore, past research found that perceived source credibility mediates the influence of political bias on an individual’s susceptibility to misinformation [55]. Thus, even though past literature found that conservatives tend to gravitate toward fake news more than liberals [28], higher NFCC in conservatives may motivate them to place higher value on perceived credible news sources because of NFCC’s desire for order and consistency, implying that NFCC may disrupt the influence of misinformation. These findings contradict H_3b_ because NFCC increased medical trust among those who preferred conservative news sources.

Moreover, among Black respondents, we found that those who preferred factual news sources were more likely to trust medical scientists. On the other hand, political bias of news sources and dispositional traits were not associated with medical trust levels. With factuality of news sources being a strong indicator of trust in medical scientists, this finding is consistent with the past literature because misinformation may exacerbate pre-existing sentiments of uncertainty toward vaccines and medical institutions that are a result of ongoing and past structural racism [17].

Although past research has examined information assessment traits on misinformation susceptibility, this study incorporated variables measuring news consumption habits. Results from our analysis suggest that it may be prudent for public health officials to account for differences in information assessment traits when designing and targeting public health campaigns and messages, rather than exclusively relying upon political groupings or other attributes used to engage in education and outreach. Moreover, by taking differences in information assessment traits into account (for example, making research results more digestible without explanations that could be perceived as condescending), it may be possible to mitigate anti-science sentiments that are driven by a perception of elitism in anti-intellectualism discourse [56]. Additionally, the observed positive associations between a preference for factual news sources and medical trust indicates that it may be worthwhile for public health anti-misinformation campaigns to directly target viewership of fringe media outlets. By directly targeting these viewers, public health officials can concentrate limited resources on audiences that are susceptible to alternative false narratives. This is particularly important as low-credibility news outlets are often the source of prominent and reoccurring false narratives [57,58]. Moreover, although this study was centered on the American political and media landscape, our results could be expanded internationally as public health crises such as COVID-19 and mainstream media effects are not contained to a single country. Future studies should take into consideration possible differences in cultural, political, and media landscapes when applying these results.

Follow-up work building on the results of this study can analyze the content in news articles from low-credibility sources to inform counter-messaging campaigns while further evaluating information assessment trait preferences from a wider group of respondents. The combination of these results may help to address specific talking points more effectively within misinformation discourse and lead to tailored public health education and promotion that is also responsive to viewers with diverse political, ideological, and conscientiousness sentiment toward science, health, and wellbeing.

## 5. Limitations

There are limitations to the current study. First, liberals were overrepresented in the data, as only 33.8% of MTurk respondents were conservative. Additionally, those recruited from MTurk were Twitter users, which may not be representative of online users or the general population. Due to content across news outlets being diverse and ranging from current event reporting to opinion segments, future studies should consider the types of content the respondent reads/watches from each news source. Preference for current event reporting versus opinion talk shows may also have utility in analyzing influencers of medical trust, misinformation spread, and other behaviors with public health implications. Moreover, the main outcome variable used in the current study may be too general of a measure for assessing the full complexities associated with trust in medical institutions since there was only a single-item question regarding trust in medical scientists. In general, a single-item measurement is not as comprehensive as a multi-item measurement. Future work should consider expanding this metric in order to investigate how news consumption habits and dispositional traits influence trust across multiple healthcare professionals such as nurses, medical doctors, and health-focused institutions such as the World Health Organization. Additionally, sample means for factuality levels of news sources preferred and people’s trust in medical scientists were relatively high. We note that this is a limitation because the sample may have been biased toward individuals with these news consumption habits and levels of trust in medical scientists. We also acknowledge that recruiting people who have rather low trust in medical institutions (and institutions in general) may be challenging for research conducted by universities due to anti-intellectualism being a prevalent sentiment amongst those with low trust in institutions.

## 6. Conclusions

Although the political slant of an individual’s preferred news sources may influence one’s trust in medical scientists, the factuality of the news sources and one’s ability to evaluate information were shown to be better indicators. This implies that, even though public health issues often become politicized, it is necessary that public health officials do not rely on political groupings and instead take into account differences in information assessment traits and target narratives from fringe media outlets when combatting misinformation.

## Figures and Tables

**Figure 1 ijerph-20-05842-f001:**
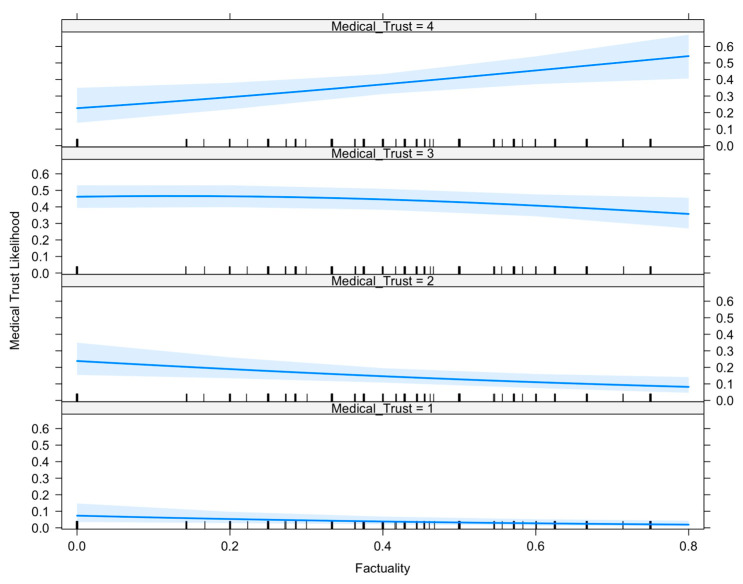
Effect of factuality on medical trust.

**Table 1 ijerph-20-05842-t001:** Classification of news sources.

	Classification	News Outlet
Political Classification	Left	Vox, MSNBC, Huffington Post, CNN
	Left-center	NYT, Washington Post, NPR, Politico, USA, ABC, CBS, NBC, BBC, PBS, Buzzfeed
	Least biased	Economist, Local
	Right-center	WSJ, Drudge
	Extreme right	Blaze, Breitbart, FOX News, Newsmax
Factuality Classification	Very high factuality	NPR
	High factuality	NYT, Economist, Politico, USA, ABC, CBS, NBC, BBC, PBS, Local
	Mostly factual	Buzzfeed, Washington Post, WSJ, Vox
	Mixed factuality	Huffington Post, Breitbart, Newsmax, Blaze, Drudge, MSNBC, CNN, FOX News

**Table 2 ijerph-20-05842-t002:** Descriptive of data.

Variable	Type	%	Count	Variable	Type	%	Count
*Gender*	Male	60.49%	519	*Race*	White	76.22%	654
	Female	37.88%	325		Black	13.87%	119
	N/A	1.63%	14		Asian	4.43%	38
					Hispanic	2.33%	20
					Other	3.15%	27
*Medical trust* *Mean = 3.106* *SD = 0.91*	1	5.94%	51	*Age*	Mean	37.26	
	2	19.11%	164		SD	10.22	
	3	33.33%	286				
	4	41.61%	357				
*CRT*	Mean	1.135		*Conscientiousness*	Mean	3.418	
	SD	1.204			SD	0.646	
*NFCC*	Mean	4.345		*Openness*	Mean	3.418	
	SD	0.815			SD	0.604	

**Table 3 ijerph-20-05842-t003:** Association of medical trust with all independent variables.

IV	Log Odds	Odds Ratio	Std. Error	T Value	*p*-Value
Factuality	1.6609	5.2639	0.4682	3.5477	0.0003
Liberal news bias	1.3230	3.7546	0.5386	2.4565	0.0140
Conservative news bias	0.1823	1.2000	0.5827	0.3129	0.7543
Conscientiousness	0.1743	1.1904	0.1318	1.3221	0.1861
CRT	0.1596	1.1730	0.0615	2.5934	0.0095
NFCC	0.1953	1.2157	0.0926	2.1094	0.0349
Openness	0.4832	1.6213	0.1485	3.2543	0.0011

Critical reflective thinking (CRT); Need for cognitive closure (NFCC).

**Table 4 ijerph-20-05842-t004:** Association of medical trust with various variables.

**Association of medical trust with conscientiousness and liberally biased news (holding factuality constant)**					
IV	Log Odds	Odds Ratio	Std. Error	T value	*p*-Value
Conscientiousness	0.6077	1.836	0.1713	3.549	<0.0001
Liberal news bias	1.241	3.459	1.144	1.084	0.2782
**A** **ssociation of medical trust with factuality, openness, CRT, NFFC, and conscientiousness (holding liberal bias news preference constant)**					
IV	Log Odds	Odds Ratio	Std. Error	T value	*p*-Value
Factuality	1.695	5.448	0.6685	2.536	0.0112
Openness	0.4775	1.612	0.2587	1.846	0.0650
CRT	0.2864	1.332	0.1141	2.51	0.0121
NFCC	0.02547	1.026	0.1741	0.1463	0.8837
Conscientiousness	0.2047	1.227	0.2327	0.8799	0.3789
**Association of medical trust with conscientiousness, factuality, NFCC, CRT, and openness (holding conservative bias news preference constant)**					
IV	Log Odds	Odds Ratio	Std. Error	T value	*p*-Value
Conscientiousness	0.1512	1.1632	0.2837	0.5329	0.5941
Factuality	1.606	4.9825	0.7947	2.021	0.0433
NFCC	0.3941	1.4831	0.1837	2.145	0.03192
CRT	−0.03173	0.9688	0.1328	−0.2389	0.8112
Openness	0.3055	1.3573	0.3036	1.006	0.3142
**Association of medical trust among black respondents**					
IV	Log Odds	Odds Ratio	Std. Error	T value	*p*-Value
Factuality	2.9839	19.7645	1.3677	2.1816	0.0291
Liberal news bias	1.3490	3.8535	1.4390	0.9375	0.3485
Conservative news bias	2.1797	8.8439	1.6576	1.3150	0.1885
Conscientiousness	0.6452	1.9063	0.3905	1.6520	0.0985
NFCC	−0.0080	0.9921	0.2715	−0.0293	0.9766
CRT	0.1445	1.1555	0.1924	0.7511	0.4526
Openness	−0.2244	0.7990	0.4359	−0.5149	0.6066

Critical reflective thinking (CRT); Need for cognitive closure (NFCC).

**Table 5 ijerph-20-05842-t005:** Association of medical trust with conscientiousness and factuality.

IV	Log Odds	Odds Ratio	Std. Error	T Value	*p*-Value
Conscientiousness	0.4693	1.5989	0.2837	4.4880	<0.0001
Factuality	1.2601	3.5259	0.7947	3.8472	<0.001

## Data Availability

The datasets used and/or analyzed during the current study are available from the corresponding author on reasonable request subject to appropriate deidentification and aggregation.

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
