# Peer review of "The Influence of News Consumption Habits and Dispositional Traits on Trust in Medical Scientists"

_ijerph, 2023, doi:10.3390/ijerph20105842_

Round 1

Reviewer 1 Report

It is a very interesting paper and it attracts my attention to explore the Influence of News Consumption Habits and Dispositional Traits on Trust in Medical Scientists.

Overall, it is a great effort and I want to suggest few changes in the following sections

1.      Introduction:

·        It would be better to explain news consumption in the introduction section and support the study by relevant theories/past research papers.

·        Further dispositional traits were not even once explained in the paper nor defined and linked with relevant hypothesis

·        Relevant literature is not cited and how the researcher came up with this relationship is lacking. No conceptual framework was shared/discussed

2.      Material and Methods

·        It is also unclear that how the sample size was finalized and Initial screening of data is missing

·        Type of study with sampling techniques needed to be mentioned

Reviewer 2 Report

This paper deals with trust in medical scientists and aims to assess the role of two major predictors in particular: media influence and dispositional traits. The topic is interesting and highly significant, and the text is by and large well-written. I have some issues with its “presentational mode” (modelling and presentation of findings), however, and below are some objections and comments that hopefully may contribute to ease the accessibility of the manuscript and also to assess its contribution to the literature.   

 ·         Page 1: The abstract should include more information on data and modelling, and less of detailed findings.

 ·         A general note on media terminology: I suggest avoiding the term “news diets”, which is not a generic term in media research. “News consumption habits”, as used in the title, is much better. Also, one does not “consume news sources”, one prefer news sources or one consume news content, I would say.

 ·         Introduction (section 1.1) is solid, but the presentation of the underlying model and the findings (from section 1.2 onwards) seems to me a little unsystematic and selective regarding which subgroup interactions are presented and how. I suggest trying to present the underlying ideas about the relationships between the predictors in a heuristic visual model, maybe also formulate some hypotheses based on previous literature.

 ·         The results section includes many different tables, almost like a research report. Would it be possible to condense the results into one or two large table(s), for instance presenting the models block-wise? Also, I miss a regression analysis of the full model, with adjusted ORs for all predictors simultaneously.

 ·         The context of the study is strongly biased towards USA (including the bulk of the literature cited). While it is always good to have a study firmly contextualised, I miss more reflection on the international significance of this study. How can American research and findings like these be generalised to other countries? What are the “lessons” for IJERPH-readers outside the US?  

 ·         The levels of trust and also the factuality of the chosen news sources is generally quite high. While this does not in any way indicate that the data set is badly designed to address the research problem as such, it may point to a kind of “limitation” of the study, as the democratic challenges related to politicisation of public health, development of conspiracy theories, misinformation (including manipulating, and being manipulated by, the media), mistrust in authorities etc may be difficult to evaluate fully in quantitative “public opinion”-like studies. Any thoughts on this? I guess I'm curious about the validity of the trust and media exposure measures, especially the extent to which they capture the distinguishing and polarising properties of the underlying phenomena.  

 ·         The conclusion is interesting, but includes a discussion that really belongs at the end of the discussion section. Conclusion should basically refer to the major empirical findings and their most obvious political implications. 

Reviewer 4 Report

The authors have done a good work but they may need to change the title to: Influence of News Consumption Habits and Dispositional Traits on Trust in Medical Scientists. 

Other sources should be used to buttress the assertion in paragraph 2, instead of citation [10] appearing consecutively.   

The sampling techniques employed in data collection from the survey respondents are not captured in the method section.

Round 2

Reviewer 2 Report

The authors have handled my objections satisfactorily.